# Cumulative semantic interference is blind to morphological complexity and originates at the conceptual level

Anna-Lisa Döring[1]*, Rasha Abdel Rahman[1], Pienie Zwitserlood[2], Antje Lorenz[1]

1 Department of Neurocognitive Psychology, Humboldt-Universität zu Berlin, Berlin, Germany,
2 Department of Psychology and Otto Creutzfeldt Center for Cognitive and Behavioral Neuroscience, Westfälische Wilhelms-Universität Münster, Münster, Germany

* anna-lisa.doering@hu-berlin.de

**Data Availability Statement:** The data underlying the results presented in the study are available from https://osf.io/qzynk/.

## Abstract

When naming a sequence of pictures of the same semantic category (e.g., *furniture*), response latencies systematically increase with each named category member. This cumulative semantic interference effect has become a popular tool to investigate the cognitive architecture of language production. However, not all processes underlying the effect itself are fully understood, including the question where the effect originates from. While some researchers assume the interface of the conceptual and lexical level as its origin, others suggest the conceptual-semantic level. The latter assumption follows from the observation that cumulative effects, namely cumulative facilitation, can also be observed in purely conceptual-semantic tasks. Another unanswered question is whether cumulative interference is affected by the morphological complexity of the experimental targets. In two experiments with the same participants and the same material, we investigated both of these issues. Experiment 1, a continuous picture naming task, investigated whether morphologically complex nouns (e.g., *kitchen table*) elicit identical levels of cumulative interference to morphologically simple nouns (e.g., *table*). Our results show this to be the case, indicating that cumulative interference is unaffected by lexical information such as morphological complexity. In Experiment 2, participants classified the same target objects as either man-made or natural. As expected, we observed cumulative facilitation. A separate analysis showed that this facilitation effect can be predicted by the individuals' effect sizes of cumulative interference, suggesting a strong functional link between the two effects. Our results thus point to a conceptual-semantic origin of cumulative semantic interference.

## Introduction

At the heart of effective language production is the lexical selection process, namely the selection of the lexical representation that best expresses the meaning of the preverbal, conceptual message. This process has been intensively investigated, mostly by picture naming studies that systematically manipulated the semantic context within which a target picture appeared. This

**Funding:** This research was entirely funded by the German Research Council (LO 2182/1-2 – granted to A.L.). There was no additional external funding received for this study. The funders had no role in study design, data collection and analysis, decision to publish, or preparation of the manuscript.

**Competing interests:** The authors have declared that no competing interests exist.

manipulation is based on the following two assumptions that are shared among most researchers in the field: 1. lexical access in speech production is semantically driven, meaning that a speaker must first activate the semantic representation of the to-be-named target (at least to a minimal degree) before lexical access is initiated, and 2. semantically related lexical entries are initially co-activated via spreading activation at the conceptual level [e.g., 1] before the target entry is selected from among the co-activated items at the lexical level [e.g., 2–10]. Regarding the latter, it is assumed that the level of co-activation on the conceptual level is directly modulated by the degree to which the concepts are related to the target concept, with stronger co-activation for closely related concepts.

Different naming paradigms have been used to investigate lexical-semantic encoding, starting with the picture-word interference (PWI) paradigm [11–14, see 15 for a recent overview], followed by the blocked-cyclic naming task [e.g., 16–20]. More recently, the continuous picture naming paradigm with its robust cumulative semantic interference (CSI) effect has become increasingly popular [21, 22]. It has been used to investigate lexical access in bilinguals [23], semantic integration of newly acquired words [24], lexical access in a social settings [25, 26], the lexical representation of compounds [27] or whether or not lexical selection is a competitive process [e.g., 22, 28, 29]. Despite this multitude of studies using CSI as a tool to investigate different research questions, not all underlying processes of the effect itself are fully understood. It is, for example, still a matter of debate where CSI originates [e.g., 3, 9, 22, 29, 30]. It is also still unknown if and how CSI effect is affected by lexical variables such as the morphological structure of the stimuli used in the naming task. The current study aims to address both of these issues.

## The origin of cumulative semantic interference

In the continuous picture naming paradigm, members of different semantic categories (e.g., *desk*, *chair*, *shelf*, *bed*, *wardrobe* for the category *furniture*) are presented for naming in a seemingly random order, separated by 2 to 8 unrelated objects [filler items or members of other categories; e.g., [22]). Participants' naming latencies within each semantic category systematically increase in a linear fashion with each ordinal position, that is, as a function of previously named objects of the same category. This CSI effect is independent of the number of intervening unrelated items [22, 28, 30–32; but see 33] and survives multiple repetition cycles [27, 28, 34]. All existing models agree that the locus of cumulative interference, that is, the level at which it comes into effect and behavioural consequences arise, is the lexical level [3, 9, 22, 28–30]. Here, CSI is interpreted in terms of increasing difficulty to select a target's lexical representation—from now on called "lemma" [6]—amongst a group of co-activated lemma representations. However, activation within the lexical system is short-lived [e.g., PWI paradigm; [12] and thus not suitable to explain the longevity and accumulating nature of CSI. Therefore, different learning mechanism have been proposed, where structural changes to the system are responsible for the persistence of the effect. The level at which these occur, meaning the level at which cumulative interference actually has its origin (as opposed to its locus, mentioned above), is still a matter of debate.

Howard and colleagues [22] and Oppenheim and colleagues [29] both locate the origin at the interface of the conceptual and lexical level, but differ with respect to the underlying learning mechanism. Howard et al. assume that producing a word strengthens the connection between the concept and its lemma entry, thus priming its future activation. A subsequently presented picture of a member from the same category activates semantically related concepts via spreading activation [1,4–7, 9, 10], including the previously named category member. Due to the now strengthened connection between the concept and lemma, its lemma is more

strongly activated than previously unnamed objects, making it a strong competitor in the lexical selection process of the to-be-named object. As each additional category member adds to the cohort of competing lexical items, the number of strong competitors systematically increases, resulting in accumulating interference [22]. While Oppenheim and colleagues [29] also localise the origin of the CSI effect at the conceptual-lexical interface, they explain cumulative interference without competitive lexical selection. in their view, incremental learning not only entails the reinforcement of connections between the conceptual and lexical entry of the named target but also the weakening of connections between concepts and lexical representations of co-activated non-targets. By means of computational modelling, Oppenheim et al., [29] showed that this error-driven learning suffices to explain the accumulation of interference without assuming a competitive lexical selection process [see also 28].

In contrast to these accounts, others argue for a purely conceptual origin of cumulative interference [9, 30]. Belke [30] proposes a learning mechanism at the conceptual level, where the links between a target's lexical concept (a unitary conceptual representation node) and its semantic features are strengthened after the target's lexical concept has been selected [30; for a model of lexical-semantic memory of this kind, see, e.g., 35]. When subsequently trying to name a semantic relative, these strengthened links will result in strong co-activation of the lexical concept named earlier and its lemma, causing competition during lexical selection of the to-be-named relative. The rationale is identical to that of Howard et al. [22] in that each additionally named member of a category will increase the competition during lexical selection, resulting in accumulating interference. Belke supports her claim about the conceptual origin by demonstrating that cumulative context effects can also be observed in purely semantic tasks that do not (necessarily) involve the lexical level. When participants classified, via button-press, objects of different semantic categories as either man-made or natural, cumulative facilitation was observed instead of interference: participants' response latencies systematically decreased within semantic categories. Belke assumes that the repeated activation of semantic features related to either man-made or natural entities of a certain semantic category induces accumulating activation at the conceptual level, rendering the man-made or natural distinction within these semantic categories increasingly easier. As the locus of the effect is identical to its origin, namely the conceptual level, facilitation instead of interference is observed. In addition, Belke reports that semantic facilitation and interference influence one another in an experiment including both tasks, picture naming and semantic classification [30, Exp. 5]. As the classification task only requires conceptual processing, Belke argues for a common conceptual origin of the two effects. While this learning mechanism had not been computationally implemented, Roelofs [9] provided a computational simulation of a similar account. Here, the learning mechanism at the conceptual level was implemented by means of a temporary bias, which not only successfully simulated cumulative interference in naming but also the cumulative facilitation reported by Belke [30].

The models assuming the origin of cumulative interference at the lexical-semantic interface were designed to accommodate speech production data, and thus do not provide explicit explanation for cumulative facilitation found in semantic classification [22, 29]. However, if we assume that the classification task does not entail activation of lexical information, a learning mechanism at the interface between the conceptual and lexical level would not seem to be able to explain cumulative facilitation observed in the classification task. Thus, further studying cumulative facilitation seems a good way forward when investigating the origin of cumulative context effects in speaking. The first aim of the current study is therefore to get more comprehensive understanding of cumulative facilitation and its similarities and difference to cumulative interference.

## Cumulative semantic interference and morphological complexity

To make informed predictions when using cumulative interference as a tool to test theories of the cognitive architecture in language production, it is also essential to identify the conditions under which it arises, and which factors modulate the effect. While plenty of studies have systematically explored the influence of different distractor-target types on semantic context effects in the PWI paradigm [7, 36–45], much less is known about their influence on cumulative interference in the continuous paradigm. Thus far, studies have shown that cumulative interference can be observed for targets that are only associatively but not categorically related [45] and that more closely related category members induced greater cumulative interference than more distant ones [34]. However, while many continuous naming studies included a mixture of simple nouns, such as *table* or *shelf*, and morphologically complex noun-noun compounds, like *woodworm* and *bookshelf* [22, 25, 30, 31, 34, 45], it is still unclear whether morphologically complex compounds induce identical CSI to their morphologically simple noun counterparts.

Given that cumulative interference is clearly semantically driven, one could argue that morphology is unlikely to be relevant at all. However, interference is assumed to come into effect at the lexical level, more specifically at the lemma level [e.g., 6, but see 4], and recent empirical evidence suggests that compounds and simple nouns may not be represented in the same way at this level. While simple nouns (e.g., *shelf*) are assumed to have a single entry at the lemma level [6], studies on compounds (e.g., *bookshelf*) suggest that they may have multiple lemma representations [27, 46, 47], namely morpheme-sized lemma entries (*book* and *shelf)* that complement the holistic compound lemma [for evidence from neuropsychological studies, see e.g., 46, 48, 49]. If compounds and simple nouns are differently represented on the level where cumulative interference is said to come into effect, different activation patterns might lead to different patterns of cumulative interference [for contrasting evidence, see e.g., 43, 50].

While PWI studies report identical interference for compound distractor-target pairs (wooden spoon—bread knife; original materials in German) and simple noun distractor-target pairs [spoon—knife; 43, Exp. 2], research suggests that effects in PWI are not necessarily transferable to the continuous paradigm [for recent discussions, see e.g., 3, 27]. The only study that systematically manipulated morphological complexity in a continuous picture naming paradigm is our recently published study that investigated the representation of German compounds in speech production [27]. In this study, participants named pictures in a compound and a simple noun condition. Category membership of compound targets was established through the compounds'first constituents (category animals: ***dog*** *lead*, ***zebra*** *crossing*, ***pony*** *tail*, ***mouse*** *trap*, ***cat*** *litter*), while the compounds themselves were not semantically related. The simple noun, control condition consisted of pictures depicting the compounds' first constituents (*dog*, *zebra*, *pony*, *mouse*, *cat*). We observed cumulative interference in the simple noun as well as the compound condition, indicating that the semantic relationship between the compounds' first constituents influenced compound production. As this suggests activation of constituent lemmas during compound production, the results support the multiple-lemma representation account of compounds [46]. Importantly, we observed significantly weaker interference for compounds than simple nouns. While one could take this as evidence for the influence of morphological complexity on CSI, it is important to remember that the interference in the compound condition was only induced by the compounds' first constituents [27]. In German, a compound's grammatical features are determined by its second constituent (i.e., its head), which also carries the bulk of the meaning of a semantically transparent compound (e.g., Zahnbürste [toothbrush]), while the second constituent, the modifier, merely provides further specification [i.e., a toothbrush is a type of brush, used for teeth; cf. 51, 52].

Consequently, any effects solely related to the modifier constituents of compound targets are likely to be weaker than those related to either the head, or the compound as a whole [see 27]. Thus, from these results we cannot infer whether semantically related compounds (*bookshelf*, *canopy bed*, *arm chair*) induce identical levels of CSI to their semantically related simple noun counterparts (*shelf*, *bed*, *chair*). The current study aims to answer this question.

## The current study

The overall purpose of this study is to gain a more comprehensive understanding of the cumulative semantic interference effect found in the continuous picture naming paradigm. We conducted two experiments with the same group of participants and the same visual stimuli.

Experiment 1 used a continuous picture naming task designed to investigate whether the CSI effect differs for morphologically complex noun-noun compounds (*bookshelf*, *kitchen table*) and their corresponding morphologically simple head nouns (*shelf*, *table*). The picture stimuli, belonging to different semantic categories (e.g., furniture, animals, clothing. . .), were selected in such a way that they could equally well be named with either a simple noun or a compound noun (e.g, *shelf* vs. *bookshelf*). In a familiarisation phase, half of the participants learned that 50% of the pictures correspond to compound names (*bookshelf*) and 50% to simple noun names (*cup)*, while the other half of the participants learned it the other way around (*shelf* and *tea cup*). Based on previous studies [22, 30], we expected to find robust cumulative semantic interference in the simple noun condition, reflected by a linear increase in naming latencies with each ordinal position, as a function of previously named pictures from the same semantic category. For the compound condition we also expected cumulative interference. However, we predicted that interference might be weaker for compounds than simple nouns as multiple lemmas (and concepts) become activated [*bookshelf*, *book*, and *shelf*, e.g., 27; but see 43]. This might lead to overall weaker activation levels, as some activation might dissipate across the semantic network via the activated concept of the compound's first constituent (*book*). As the first constituent was not part of the same semantic category as the compound itself (i.e., furniture), its reciprocal activation (from lemma to concept) would co-activate other members of its semantic category (i.e., journal, magazine . . .), which could result in weaker accumulating interference within the category of the compound. In case of an observed difference in interference between the two word types, we wanted to ascertain that this would not be due to differences in semantic similarity between category members in the simple noun or the compound condition. Thus, we included a measure for semantic similarity into the analysis to control for this semantic factor.

Experiment 2 employed a semantic classification task, in which the same participants saw the picture stimuli from Experiment 1 in the exact same order and were instructed to classify the depicted objects as either man-made or natural (via button-press). The aim of this experiment was twofold. First, we wanted to replicate cumulative facilitation which, thus far, has only been reported once in a manual task [30] and once in a verbal classification task [53]. At the same time, we wanted to gain a better understanding of cumulative facilitation by investigating whether it survives multiple repetitions and whether it is modulated by semantic similarity. This would provide a more comprehensive profile of cumulative facilitation concerning its commonalities and differences to cumulative interference.

Second, we wanted to investigate whether one semantic context effect can be used to predict the other, which would point towards a causal link between the two effects and thus corroborate a conceptual origin of cumulative semantic interference in the picture naming task [30]. As the classification task always followed the naming task, we used the interference effect found in Exp 1 to predict the facilitation effect in Exp 2.

We included Word type as a predictor in the analysis of the classification task to control for the possibility that participants activated different lexical concepts / a different set of conceptual features upon seeing the pictures as a result of previous naming them with different labels (e.g., either as *kitchen table* or *table*; note that each participant named half of the targets with a simple word, the other half with a compound). However, even if this were the case, we expected identical levels of facilitation, independent of previous simple-word and compound naming. The task requires participants to activate nodes identifying the object as either man-made or natural, which should be identical for lexical concepts such as *KITCHEN TABLE* and *TABLE.* As activation levels should also be comparable for both concepts with regards to man-made or natural nodes, we expected similar facilitation for both previously used word types.

## Experiment 1

### Material and methods

**Participants.**    Thirty-six native speakers of German (20 female, 16 male) between the age of 18 and 35 (mean 26.7 years) were included in the analysis. Due to technical problems, three participants had to be excluded and replaced. The sample size was chosen following a power analysis [simr package, 54]. Based on a previous experiment using linear mixed-effects models to analyse log-transformed reaction times from a continuous picture naming study with simple-noun targets [25], we predicted an effect size (*b)* of about 0.04 for the interference effect in the simple noun (control) condition for five presentations (naming cycles 1–5). To account for the possibility of a smaller interference effect in the compound condition, we used a *b* of 0.025 when simulating the outcome of the anticipated model with 1000 iterations for the compound condition. With 36 participants we reached a power estimate of 84,6% (95% confidence interval: 82.2, 86.8) for detecting the hypothesized cumulative semantic interference in the compound condition. All participants had normal or corrected-to-normal vision and received monetary compensation or course credit for their participation. The study was approved by the local ethics committee of Humboldt-Universität zu Berlin via written consent and is in accordance with the Declaration of Helsinki. All subjects gave informed written consent.

**Materials.**    The stimuli set consisted of 140 coloured photographs of 70 man-made and 70 natural entities and their written names. The set included 90 targets, 30 filler items and 20 practice items. The targets belonged to 18 different semantic categories (e.g., clothes), with five members each. Care was taken that each photograph could be named with a simple noun (e.g., *Bluse* (blouse)) or a noun-noun compound (e.g., *Seidenbluse* (silk blouse)). To verify the suitability of our stimuli set, two pre-studies were conducted. In the first online survey, we established a measure for picture/label-fit, to control for possible differences between the two word types. Forty native speakers of German (mean age: 26.9 years; range: 18–47) took part in the survey (20 participants per picture-label pair). The participants were presented with a picture and the corresponding label in either the compound (*silk blouse*) or the simple noun (*blouse*) condition and were asked to use a six-point Likert scale (6 = perfectly/entirely; 1 = not at all) to indicate how well the label describes the picture. The results showed that the compound labels (mean rating: 5.34, *SD* = 1.0) and the corresponding simple noun labels (mean rating: 5.36, *SD* = 1.0) described the pictures equally well (*t* = 0.30, *p* = 0.77). As prior research suggests that within-category semantic similarity modulates cumulative interference [34], we conducted a second online survey to establish a measure of semantic similarity for the statistical analysis. Eighty native speakers of German (mean age: 33.5 years; range: 18–70) took part in the survey (20 participants per item pair). The participants were presented with word pairs (i.e., two members of a category) and were asked to use a six-point Likert scale (6 = very closely related; 1 = not at all related) to indicate the semantic similarity of the two items. Semantically similar

items were defined as sharing many semantic features (e.g., *apricot* and *plum*: both are fruits, can be eaten, grown on trees, have a stone, are round . . .), while items with few or no shared features were defined as semantically distant (e.g., *apricot* and *telephone*). For each material set, participants either rated two compounds (e.g., *silk blouse* and *winter coat*) or the corresponding simple nouns (e.g., *blouse* and *coat*). The results showed that the members of the 18 categories were overall perceived as closely related (mean rating overall: 4.5, *SD*:1.2). This was confirmed for both word types, but additional analyses showed that simple nouns were perceived as significantly more closely related (mean: 4.64, *SD*: 1.31, range: 3.3–5.5) than compounds (mean: 4.40, *SD*: 1.35, range: 3.0–5.1, $t = 10.67$, $p < 0.001$).

The 50 filler and practice items were semantically unrelated to the targets and care was taken to ensure that there were no morphological overlaps, meaning that no constituent appeared more than once in the stimuli set. Due to an oversight, the constituent *Hammer* (hammer) was included twice, once as the first constituent in the compound *Hammerhai* (hammerhead (shark)) and once as the second constituent in *Gummihammer* (rubber hammer). All photographs were scaled to 3.5cm x 3.5cm and had a homogenous light grey background. Appendix A lists all materials used in this experiment.

**Apparatus.** Participants were seated in a noise-cancelling booth, approx. 80 cm from the screen and approx. 30 cm from the microphone. The pictures were presented on a 19" inch screen (1280x1024), using version 17.0 01.14.14 of the software Presentation® (Neurobehavioural Systems, Inc, www.neurobs.com) and response times were registered by a voice-key (self-made) and a Sennheiser MKH 416 P48 microphone.

**Experimental design.** Eighteen different lists were created on the basis of a master list that contained 90 slots for the target words and 30 slots for the filler items. The five slots for the five members of a category were separated by 2, 4, 6 or 8 intervening items [lag value; e.g., 22]. The order in which categories appeared within a list and the order of the five members within each category were unique for each list. Care was taken to keep semantically related categories apart (e.g., land animals, marine animals, and insects) to avoid participants creating superordinate categories.

The two word types (compounds, simple nouns) were presented block-wise, thus every list was split into two blocks, each containing 45 critical and 15 filler items in the compound or simple noun condition. To directly compare the processing of compounds and simple nouns, two versions of each of the 18 lists were created and presented to different participants. In version one, participants were asked to use simple nouns when naming the pictures of the first block (e.g., *Ring* (ring)) and compounds when naming the pictures of the second block (e.g., *Orangensaft* (orange juice)). In version two, participants were asked to use compounds for the first block (e.g., *Ehering* (wedding ring)) and simple nouns for the second (e.g., *Saft* (juice)). This counterbalanced order ensured that each participant saw each picture in only one word type condition. Note that participants were not explicitly informed about the two word types, and simply learned the picture labels during the familiarisation phase. Every participant was presented with five differently randomised versions of their list to enhance statistical power and to further investigate the effect of repetition on the cumulative semantic interference effect. This factor will be called *Presentation*. To ensure that participants did not have to switch between the two word type conditions, we used a blocked design.

**Procedure.** Prior to each word type condition (compounds, simple nouns), there was a familiarisation phase. Participants were presented with the 70 pictures and corresponding labels of the upcoming word type condition (45 targets, 15 filler items, 10 practice items) in a random order on the screen (one picture at a time) and were asked to remember the correct label. They moved on to the next picture by themselves and thus had as much time as needed during the familiarisation phase. In the main experimental session, a fixation cross was shown

for 500 ms at the start of each trial, followed by the picture. The picture was presented until a response was initiated or for a maximum of 2500 ms. After an inter-trial interval of 2 seconds, the next trial started. Participants were instructed to name the pictures as fast and as accurately as possible, using the label they had seen during the familiarisation phase. Naming latencies were recorded with the help of a voice-key from picture onset and the experimenter coded any voice-key or naming errors (incorrect responses, stuttering etc.).

## Analysis

The data analysis was done with R [55]. Naming latencies were analysed for target trials only. Trials in which pictures were named incorrectly or dysfluently (3.5%), in which the voice-key was triggered too late (0.9%) or in which other technical or experimenter errors occurred (1.6%) were excluded from the analysis. For the identification of outliers, we combined light a-priori screening for artefactual responses with a removal of outliers that were not within normal distribution of the final model's residuals [for more details on the procedure, see 56]. The a-priori screening resulted in an exclusion of 1.3% of the data (RT < 300 ms), while a further 2.55% were excluded after model fitting (standardised residuals > than 2.5).

The inverse-transformed reaction times were fitted with a series of linear mixed effect models [LMM; 57], using the function lmer of the R package lme4 [58] and $p$-values were computed with the lmerTest package [59]. As the reaction time data were not normally distributed, we used the Box—Cox procedure [60] implemented in the boxcox-function in the package MASS [61] to identify the most appropriate transformation. However, untransformed RTs yielded similar results. Model comparisons were performed to identify the best fitting model. Starting with a maximal model, the model was first simplified by successively removing those random effects that explained the least variance, aiming to include the maximal random effect structure which enables model convergence and does not lead to overfitting [using the rePCA function; see 62]. The fixed effect structure was then reduced by successively excluding covariates and/or interaction terms, and then compared until the simpler model explained the data significantly worse than the more complex one (significant $\chi$2 test in the anova function). The fixed structure of the initial model also included the covariates Lemma frequency and Word length. Neither of them improved model fit and were thus excluded from the analysis.

The final model used for the main analysis included main fixed effects and a three-way interaction of the predictors Word type (compound vs. simple noun), Ordinal position (five ordinal positions of members within one category) and Presentation (Presentations 1–5, i.e., first naming cycle and four repetitions with different lists) and main fixed effects and a three-way interaction of Word type, Ordinal position and Semantic similarity (for each item: mean values from semantic similarity rating). Picture fit (for each item: mean values from picture/label-fit rating) and Trial (consecutive trial number) were included as covariates. The latter was included to account for changes in the course of the experiment [i.e., trial-by-trial sequential effects, e.g., 53, 56, 63]. The random structure included random intercepts for Subjects, semantic categories and items nested within categories, random slopes for Word type for each participant, as well as random slopes for Presentation, Ordinal position, Picture fit, Semantic similarity and the interaction of Word type and Presentation for each participant (omitting correlations to facilitate convergence). The predictor Word type was contrast-coded using effect coding, while the predictor Ordinal position was contrast-coded using polynomial contrasts. Polynomial contrasts test for a linear trend in the data (among others), that is, whether the increase in response times from ordinal position 1 to 5 is linear or not. The cubic and

quadratic trends were excluded from the analysis as they did not improve model fit [for more details on contrast-coding, see 64]. The two predictors Presentation, Semantic similarity and the covariates Picture fit and Trial were centred and entered as continuous variables.

In a second analysis we included the factor Lag (number of intervening items between the category members on Ordinal position 1–5) as an additional predictor in the above-mentioned model to investigate its influence on cumulative interference. Here, only data from ordinal position 2–5 were considered because there is no lag before ordinal position 1 [e.g., 18, 26, 29]. The factor Lag was centred and then added to the above model as a fixed effect in a three-way interaction with Ordinal position and Presentation.

## Results

Table 1 contains the results of the main LMM analysis. As expected, naming latencies were significantly longer for compounds (Ø = 863.0 ms) than for simple nouns (Ø = 752.1 ms; main effect: word type). Furthermore, Picture fit significantly influenced overall naming latencies (main effect: Picture fit), with shorter naming latencies for items with higher picture-fit ratings. Crucially, the data shows a significant main effect of Ordinal position, reflecting the linear increase of naming latencies from one ordinal position to the next

**Table 1. Main model Experiment 1.** 1000/RT ~ Word type*Ordinal position *Presentation + Word type*Ordinal position*Semantic similarity + Picture fit+ Trial + (Word type*Presentation+ Ordinal position +Picture fit+ Semantic similarity||Subject) + (1|Category \ Item).

| Predictors | -1000/RT | | | |
|---|---|---|---|---|
| | Estimates | std. Error | t-value | p |
| (Intercept) | -1.35 | 0.029 | -46.73 | **<0.001** |
| Word type | 0.21 | 0.031 | 6.87 | **<0.001** |
| Ordinal position | 0.09 | 0.008 | 11.61 | **<0.001** |
| Presentation | -0.07 | 0.004 | -16.52 | **<0.001** |
| Semantic similarity | 0.08 | 0.021 | 3.65 | **<0.001** |
| Picture fit | -0.11 | 0.020 | -5.33 | **<0.001** |
| Trial | <0.001 | <0.001 | 3.14 | **0.003** |
| Word type * Ordinal position | -0.02 | 0.010 | -1.54 | 0.123 |
| Word type * Presentation | -0.02 | 0.005 | -3.32 | **0.002** |
| Ordinal position * Presentation | < -0.01 | 0.003 | -0.64 | 0.520 |
| Word type * Semantic similarity | -0.04 | 0.040 | -1.05 | 0.290 |
| Ordinal position: Semantic similarity | 0.02 | 0.010 | 2.07 | **0.039** |
| Word type * Ordinal position* Presentation | -0.01 | 0.007 | -0.92 | 0.355 |
| Word type *Ordinal position * Semantic similarity | 0.02 | 0.020 | 1.17 | 0.241 |

**Random Effects**

| | Variance | Sd |
|---|---|---|
| Subjects (Intercept) | 0.02 | 0.16 |
| Subjects (Word type) | 0.01 | 0.10 |
| Subjects (Presentation) | < 0.01 | 0.02 |
| Subjects(Ordinal Position) | < 0.01 | 0.04 |
| Subjects (Picture fit) | < 0.01 | 0.04 |
| Subjects(Semantic similarity) | < 0.01 | 0.04 |
| Subjects (Word type * Presentation) | < 0.01 | 0.03 |
| Item (Intercept) | 0.002 | 0.05 |
| Category \Item (Intercept) | <0.02 | 0.13 |
| Residuals | 0.07 | 0.26 |

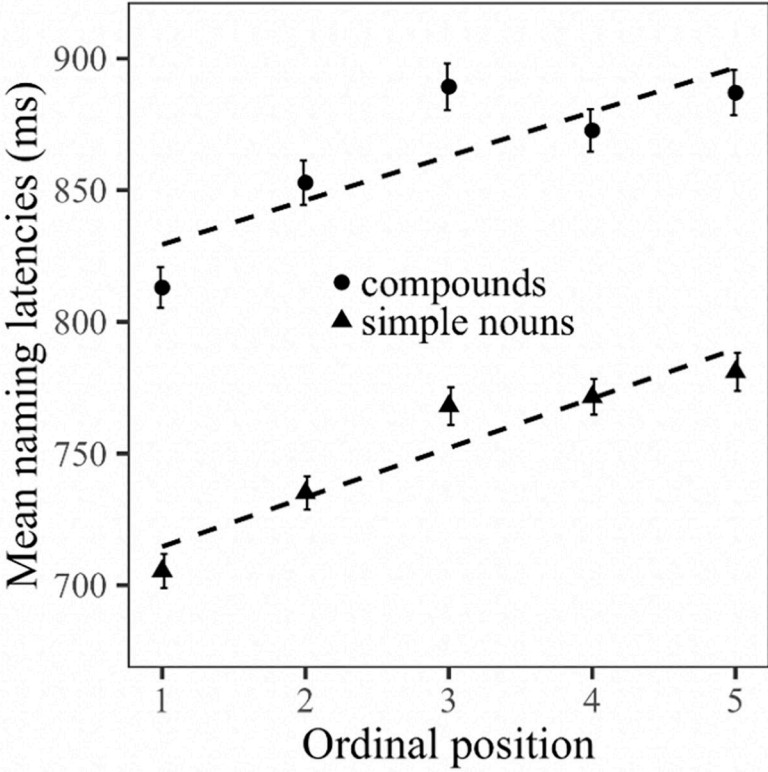

**Fig 1. Mean reaction times (naming latency) and standard error (in milliseconds) observed in Experiment 1 broken down by ordinal position and word type.**

within semantic categories. Importantly, however, the factors Ordinal position and Word type did not interact, indicating that the magnitude of the cumulative semantic interference effect (i.e., the slope of the linear increase) was identical in both word type conditions (average increase per ordinal position: compounds = 18.5 ms; simple nouns = 19.0 ms, see Fig 1). Overall, naming latencies increased throughout the experiment (main effect Trial) but decreased with each repetition (main effect Presentation), an effect that differed between the word types (interaction Word type*Presentation). A post-hoc analysis using a nested version of the same model showed that the factor Presentation had a greater influence on compound $(t = -15.84, p < 0.001)$ than on simple noun targets $(t = -12.42, p < 0.001)$. However, there was no significant interaction between Ordinal position and Presentation and no interaction between Word type, Ordinal position and Presentation, suggesting that the cumulative semantic interference effect was present for both word types in all five naming cycles. We also found a significant interaction between Ordinal position and Semantic similarity, an effect that did not differ between word type conditions (interaction Word type*Ordinal position*Semantic similarity). As illustrated in Fig 2, the closer the items of one category are related (high semantic similarity), the larger the observed interference effect (difference in naming latencies between the first and last member of each category).

The results of the second analysis including Lag as an additional predictor mirror those of the main model, so we only report the results concerning the new predictor. There was no main effect for Lag $(t = 0.86, p = 0.39)$ and only a trend for the interaction between Lag and

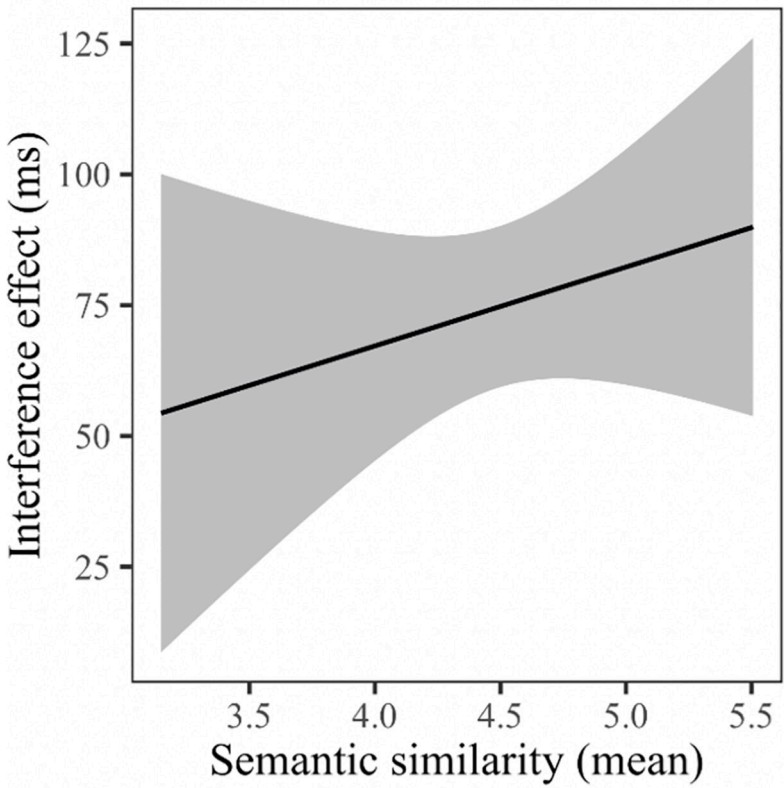

**Fig 2. Visual illustration of the interaction between the interference effect (difference between mean naming latencies on ordinal position 1 and 5 (in ms)) and the semantic similarity rating score.**

Ordinal position ($t$ = -1.84, $p$ = 0.07). However, the three-way interaction between Lag, Ordinal position and Presentation was significant ($t$ = -2.59, $p$ = 0.01), suggesting that the influence of Lag on the interference effect differed between repetitions. This was confirmed by separate post-hoc analyses of each of the five presentations. While the interference effect was independent of lags in presentations 1–4 (all p > 0.1), it was significantly influenced by Lag in presentation 5, with shorter lags inducing stronger interference than longer lags (see S2 Appendix for a visual illustration of the predicted effect). However, when collapsed over all presentations, all four levels of the factor Lag (i.e., 2,4,6,8) induced significant cumulative interference when analysed separately (Lag 2: $t$ = 3.71, $p$ < 0.001; Lag 4: $t$ = 4.13, $p$ < 0.001; Lag 6: $t$ = 2.76, $p$ < 0.01; Lag 8: $t$ = 2.62, $p$ = 0.01).

## Discussion

In Experiment 1, we observed the expected cumulative semantic interference effect [22] and found that the magnitude of the interference effect was modulated by the semantic similarity between category members. More closely related items caused greater interference, corroborating results from other continuous naming studies [34]. With regard to word type, participants took significantly longer to produce compounds than simple nouns. This expected finding can either be attributed to word-type inherent differences, such as word length and frequency, or to the fact that all compounds were subordinate-level words, which often induce longer naming latencies than basic-level simple nouns [44, 65–67]. Our key finding, however, concerns the identical slopes of increase in naming latencies from one

ordinal position to the next, for both word types. This demonstrates that cumulative semantic interference is identical for simple nouns and noun-noun compounds, meaning the effect is not influenced by the morphological complexity of the targets. We will return to this issue in the General discussion. Furthermore, the results of the main analysis showed that cumulative interference robustly survives multiple repetitions of the same items, in line with results from previous studies [27, 28, 34]. However, when investigating the influence of Lag on cumulative interference, we observed an interaction between Lag, Ordinal position and Repetition, suggesting that cumulative interference is not entirely unaffected by repetitions. Although previous studies have shown that cumulative interference is not affected by lags of less than eight intervening items between members of a category [e.g., 18, 26, 29], our results suggest that this changes when participants repeat responses several times. After multiple naming cycles of the same targets, longer lags seem to induce weaker interference than shorter ones. This suggests that general repetition priming induced by naming items multiple times [68, 69] affects the build-up of interference in the long lag- condition to a greater degree that in the short-lag condition. This is compatible with the idea that cumulative interference dissipates over time, put forward by Schnur [33]. She showed that cumulative interference does not survive multiple long lags (8–12 intervening trials), unless short lags are inserted that amplify interference within semantic categories [see Exp. 3 in 33]. In the current study, the strong and long-lived facilitation induced by repeating identical targets [68, 69] increases each time a target is named (main effect Presentation). In later presentations, this built-up facilitation affects cumulative interference at long lags (6 or 8) to a larger degree than at short lags (2 or 4), as cumulative interference dissipates over lags, and is thus more easily cancelled out by the facilitation.

## Experiment 2

Experiment 2 uses a non-verbal semantic classification task, including the identical picture stimuli as Experiment 1. Participants were instructed to indicate via button press whether a presented picture showed a natural or a man-made entity. It was designed to investigate semantic-conceptual processing of the experimental materials. We expected to replicate cumulative semantic facilitation, that is, a linear decrease of reaction times within semantic categories [30]. The aim was to better understand cumulative facilitation and its similarities and differences to cumulative interference observed in picture naming. To this end, we also investigated whether the cumulative facilitation effect could be predicted by cumulative interference, which would suggest a functional link between the two. This is important to make further inferences about the functional origin of cumulative context effects.

### Material and methods

**Participants, materials, apparatus, experimental design.** Participants, materials, apparatus and experimental lists were identical to Experiment 1. Each participant first completed Experiment 1, and, after a break of approx. 15 to 20 minutes, continued with Experiment 2. When choosing the experimental items for Experiment 1 and 2, care was taken that targets as well as filler and practice items included an equal number of natural and man-made stimuli. Furthermore, when constructing the experimental lists, we ensured that there were no more than five man-made or natural items in a row to avoid bias in the classification task. Each participant was presented with the same lists as in Experiment 1.

**Procedure.** In the experimental session, a fixation cross was presented for 500ms at the start of each trial, followed by the picture. The picture was presented until a response was initiated or for a maximum of 2500 ms. After an inter-trial interval of 2 seconds, the next trial

started. Participants were instructed to indicate via button-press whether the pictures depicted natural or man-made items. Reaction times were recorded from picture onset and incorrect responses were automatically coded.

## Analysis

The data analysis of the classification task (software, packages, contrast-coding, transformation and model selection procedure) in Experiment 2 was identical to Experiment 1, with two exceptions. First, based on previous studies [e.g., 30], we included the factor classification type (man-made vs natural) as an additional predictor into the analysis. Second, we slightly changed the a-priori screening for artefactual responses after visual inspection. In this experiment, we defined outliers as reaction times shorter than 250 ms and longer than 2000 ms. Again, only experimental trials were included in the analysis. 8.0% of the trials were excluded because participants had either incorrectly classified the items (7.5%) or did not respond within the time limit (0.5%). A further 0.5% of the trials were identified as outliers. Of the remaining data, 1.8% were excluded after model fitting (standardised residuals > 2.5), leaving 89,7% of the initial data points to be included in the analysis. The final model used for the main analysis included an interaction of the predictors Word type produced in Experiment 1 (compound vs. simple noun) and Ordinal position (five ordinal positions of category members), an interaction of Presentation (Presentations 1–5, i.e., first classification cycle and four repetitions with different lists) and Ordinal position, an interaction of Semantic similarity (rating values) and Ordinal position, a three-way interaction of Semantic similarity, Ordinal position and Presentation, an interaction of Ordinal position and Classification type (man-made vs. natural), as well as main fixed effects for all predictors, and Trial number (consecutive trial number) as a covariate. The random structure included random intercepts for Subjects, Semantic categories and Items nested under Semantic categories, random slopes per subject for the interaction of Word type and Ordinal position, as well as for Word type, Ordinal position, Presentation, Semantic similarity and Classification type (omitting correlations to facilitate convergence). Furthermore, it included random slopes for Presentation for the random factor Semantic category.

In a second analysis we included the factor Lag (number of intervening items between the category members on Ordinal position 1–5) as an additional predictor in the above-mentioned model to investigate its influence on cumulative facilitation, following the same procedure as in Experiment 1.

To test whether the facilitation observed in the classification task can be predicted by the interference observed in picture naming, we conducted a third analysis. For that, we first computed both context effects by calculating the reaction time difference (difference score in ms) between Ordinal position 1 and 5 for each subject, category and presentation for each of the tasks. Categories for which no facilitation effect or no interference effect could be computed due to missing were excluded from the analysis (24.3% of trials). The facilitation effect as dependent variable was log- transformed, while the interference effect as one of the independent variables was centred. Both were included in a linear mixed model containing Presentation (centred), Semantic similarity (mean semantic similarity for each category, centred) and Word type as additional main fixed effects as well as two-way interactions between Interference effect and Presentation, Interference effect and Semantic similarity, Interference effect and Word type, and a three-way interaction of Interference effect, Presentation and Semantic similarity. Subjects were included as random factor to account for by-subject variance. Model comparisons were performed until the best fitting model (above) was identified (following the procedure described for Experiment 1).

## Results

Table 2 shows the statistical results of the main analysis of the classification task. Overall, participants' response latencies decreased with each repetition (main effect Presentation). There was no main effect for Word type, indicating that participants took equally long to classify objects that were previously named using a compound or simple noun (mean RTs: 595.2 ms and 587.1 ms respectively). There was a main effect for Ordinal position: within semantic categories, participants' reaction times systematically decreased with each additionally classified picture (overall decrease from Ordinal position 1 to 5: 21 ms, see Fig 3 for a visual illustration). While this facilitation effect did not significantly differ as a function of word type (interaction: Word type*Ordinal position), it varied across repetitions (interaction: Ordinal position*Presentation). Separate post-hoc analyses of each of the five presentations revealed stronger facilitation (i.e., main effect for Ordinal position) in the first presentation (59ms between category member 1 and 5; $t = -5.80$, $p < 0.001$) than in presentations 3 to 5 (Rep3: 20ms; $t = -2.46$, $p = 0.019$; Rep4: 16ms; $t = -2.23$, $p = 0.026$; Rep5: 15ms; $t = -2.16$, $p = 0.037$) and no significant effect in Presentation 2 (14ms; Rep2: $t = -0.62$, $p = 0.54$, see Fig 4 for a visual illustration). Furthermore, we found a marginally significant interaction of Ordinal position and Classification type (man made—nature made). Separate post-hoc analyses for the two classification types

**Table 2. Main model Experiment 2.** -1000/RT ~ Word type*Ordinal position + Ordinal position *Presentation+ Ordinal position: Semantic similarity: Presentation + Ordinal position: Semantic similarity + Semantic similarity + Ordinal position *Classification type+ Trial+(Word type* Ordinal position +Presentation+ Semantic similarity +Classification type||Subject) + (1|Category/Item)+ (0+Presentation||Category).

| Predictors | -1000/RT | | | |
| --- | --- | --- | --- | --- |
| | *Estimates* | *std Error* | *t-value* | *p-value* |
| (Intercept) | -1.846 | 0.047 | -39.48 | <0.001 |
| Word type | 0.006 | 0.037 | 0.17 | 0.87 |
| Ordinal position | -0.044 | 0.009 | -4.88 | <0.001 |
| Presentation | -0.099 | 0.009 | -10.90 | <0.001 |
| Semantic similarity | -0.032 | 0018 | -1.80 | 0.073 |
| Classification type | 0.007 | 0.035 | -0.19 | 0.849 |
| Trial | < -0.000 | <0.000 | -1.794 | 0.080 |
| Word type * Ordinal position | 0.024 | 0.016 | 1.50 | 0.140 |
| Ordinal position * Presentation | 0.012 | 0.004 | 2.69 | 0.007 |
| Ordinal position* Semantic similarity | 0.025 | 0.013 | 1.92 | 0.06 |
| Ordinal position* Classification type | 0.022 | 0.013 | 1.75 | 0.081 |
| Ordinal position* Presentation * Semantic similarity | 0.022 | 0.009 | 2.54 | 0.011 |

**Random Effects**

| | Variance | Sd |
| --- | --- | --- |
| Subject (Intercept) | 0.06 | 0.1 |
| Subjects (Word type) | 0.02 | 0.13 |
| Subjects (Presentation) | < 0.01 | 0.05 |
| Subjects (Ordinal Position) | < 0.01 | 0.04 |
| Subjects (Semantic similarity) | < 0.01 | 0.04 |
| Subjects (Word type * Ordinal position) | < 0.01 | 0.05 |
| Subjects (Classification type) | 0.01 | 0.12 |
| Category (Intercept) | < 0.01 | 0.08 |
| Category \ Item (Intercept) | < 0.01 | 0.05 |
| Category (Presentation) | < 0.01 | 0.02 |
| Residuals | 0.11 | 0.34 |

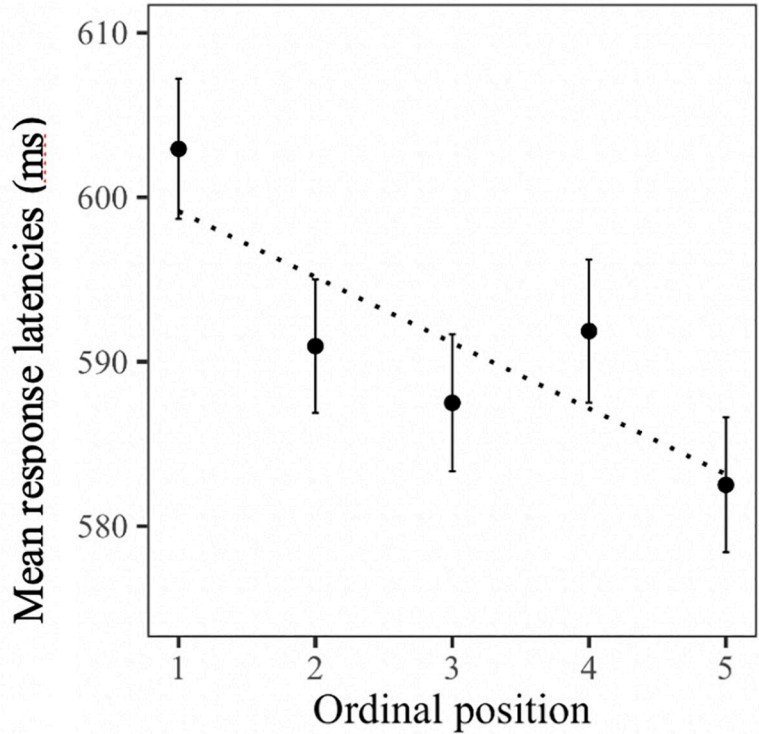

**Fig 3. Mean reaction times (response latency) and standard error (in milliseconds) observed in Experiment 2 broken down by ordinal position.**

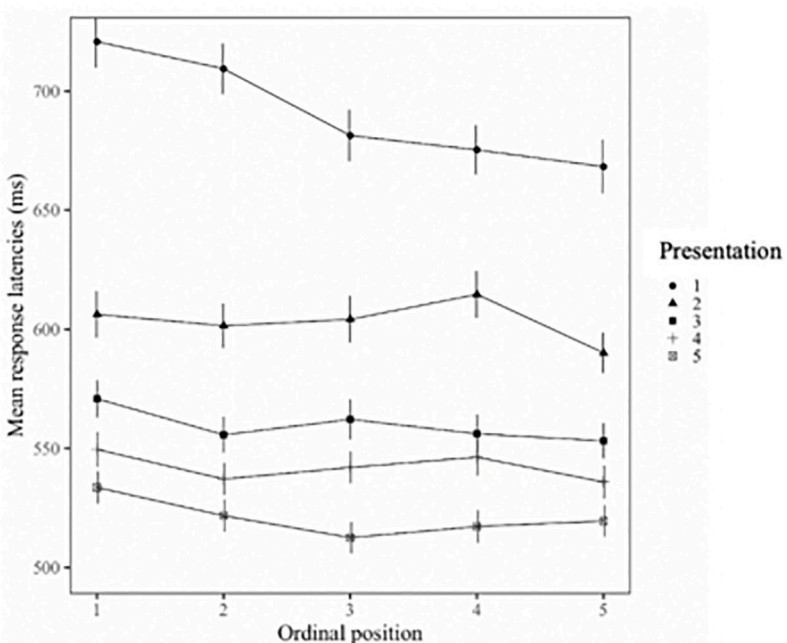

**Fig 4. Visual representation of the facilitation effect in all five presentations.**

revealed that natural items induced stronger facilitation ($t$ = -6.38, $p$ < 0.001) than man-made ones ($t$ = -3.77, $p$ < 0.001). In the main analysis, we only found a marginally significant effect for the influence of semantic similarity on the ordinal position effect (Interaction: Ordinal position*Semantic similarity) but also found that this was influenced by repetition (interaction: Ordinal position*Presentation*Semantic similarity). The above-mentioned separate analyses of the five presentations showed that semantic similarity did not influence cumulative facilitation in the first three presentations (all interaction terms p> 0.05) but more strongly related items induced weaker facilitation than more weakly related ones in the last two presentations (Presentation 4: $t$ = 2.13, $p$ = 0.03; Presentation 5: $t$ = 2.16, $p$ = 0.03; see S2 Fig in S2 Appendix for a visual illustration).

The second analysis including Lag as an additional predictor revealed a main effect for Lag (t = 4.83, p <0001), with short lags predicting faster response times, but no significant interaction with Ordinal position (t = -0.95, p = 0.35) nor with Ordinal position and Presentation (t = -0.23, p = 0.822).

S1 Table in S2 Appendix contains the results of the LMM-analysis for the third analysis of the classification task data, in which we use the interference effect (Exp. 1) as a predictor for the facilitation effect (Exp. 2). The results show a main effect for the factor Interference effect ($t$ = 2.36, $p$ = 0.019), indicating that strong interference predicts weak facilitation. There was also a main effect of Presentation ($t$ = 2.94, $p$ = 0.003) but no interaction of the two ($t$ = -0.69, $p$ = 0.49). Furthermore, we found no main effect for Word type but a significant interaction of Interference effect and Word type, suggesting that the Interference effect does not predict the facilitation effect equally for the two word types. Separate post-hoc analyses for the two word types confirmed that the interference effect can only predict the facilitation effect of those items that were named as simple nouns (t = 2.74, p = 0.006) but not for compounds (t = -0.91, p = 0.36), as visualised in Fig 5. While there was no main effect for Semantic similarity, both the two-way interaction with Interference effect ($t$ = -2.08, $p$ = 0.038) and the three-way interaction with Interference effect and Presentation ($t$ = 3.95, $p$ < 0.001) were significant. The direction of the effects suggests that strong interference predicts weak facilitation for more loosely related items, but not for closely related ones, and that this prediction becomes weaker with each presentation. However, as the post-hoc analyses above showed that only the facilitation of the simple nouns can be predicted from the interference effect, it seems more appropriate to only take the simple-noun data into account when investigating the influence of semantic similarity on the ability to predict the facilitation from interference. Therefore, in an additional analysis, only the simple noun data was analysed. We found a significant interaction between Interference effect and Semantic similarity ($t$ = -2.03, $p$ = 0.04) and the visual inspection of the interaction confirms that a strong interference effect predicts a weak facilitation effect only for weakly-related category members but not for closely related ones (see Fig 6). The three-way interaction between Interference effect, Semantic similarity and Presentation was not significant ($t$ = 0.50, $p$ = 0.62).

## Discussion

In Experiment 2, we replicated the cumulative facilitation effect reported by Belke [30]. Participants' response latencies systematically decreased within semantic categories with each classified picture (about 5ms from one picture to the next), independent of whether the pictures were named with a compound or a simple noun in the preceding picture naming task. However, we also found that this cumulative facilitation effect was influenced by repetition. While we observed a facilitation effect of nearly 15 ms (from one category member to the next) in the first classification cycles, in the subsequent four repetitions (Presentations 2–5) we observed

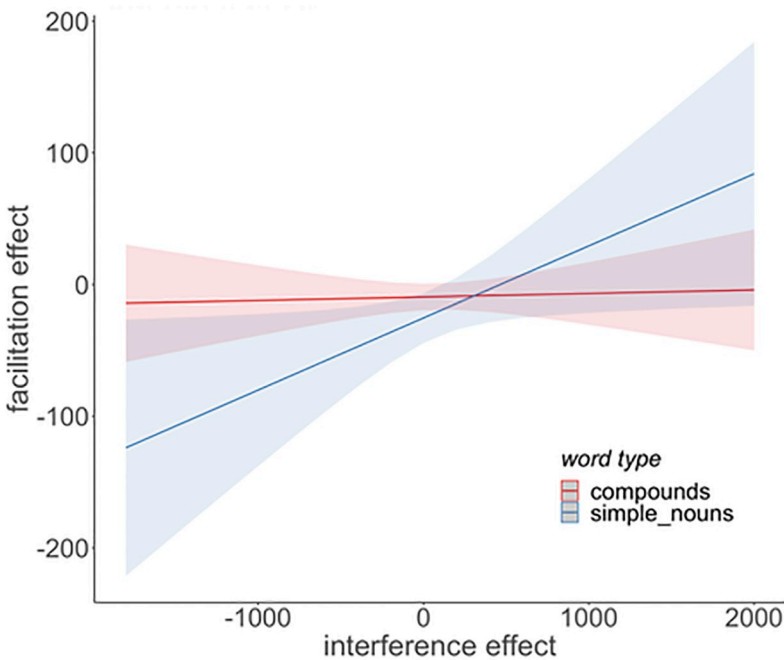

**Fig 5. Predicted facilitation effect of Experiment 2 (in ms) by the interference effect observed in Experiment 1 (in ms), broken down by word type condition.** Interference can only predict facilitation of simple nouns but not of compounds.

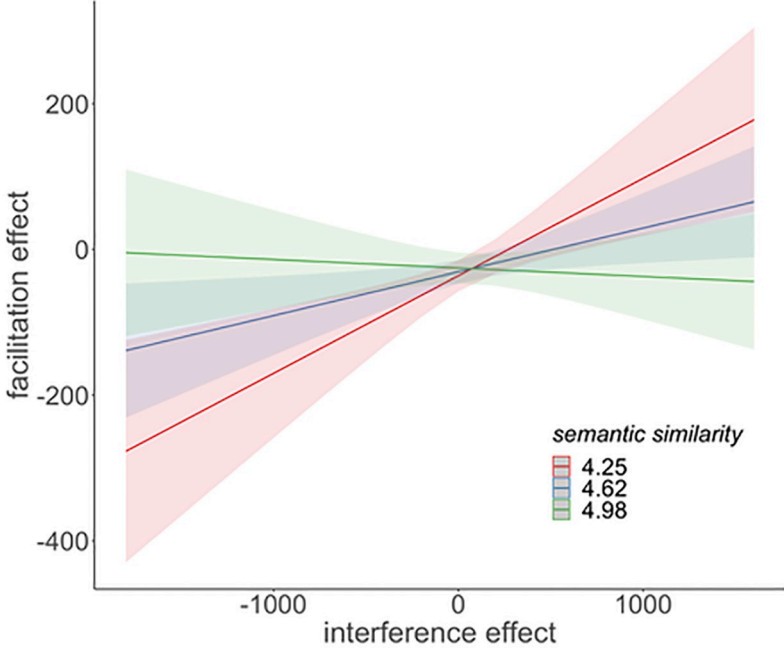

**Fig 6. Depiction of the interaction of Interference effect and Semantic similarity found in the analysis for simple nouns.** A strong interference effect (x-axis, in ms) predicts a weak facilitation effect (y-axis) but only for items that are less-closely related.

much weaker ones of 4 to 5ms. In fact, Presentation 2 showed no significant facilitation, although response latencies between the first and last category members decreased nearly as much as in Presentations 3–5. To better understand the effect found in Presentation 2, we re-ran its analysis, this time including the cubic and quadratic trends in addition to the linear one. The results revealed a significant cubic trend ($t$ = -2.26, $p$ = 0.02), mirroring the visual effect depicted in Fig 4. Overall, these results show that cumulative facilitation seems to be more vulnerable to repetitions than cumulative interference, which extends previous findings on cumulative facilitation [30, 70] and provides additional insights into the characteristics of this effect. We will discuss this further in the General Discussion. Furthermore, we found that semantic similarity does not modulate cumulative facilitation in the same way as cumulative interference, with less closely related items inducing greater facilitation after several repetitions. This, again, shows the large influence of repetition on cumulative facilitation and all its modulating factors. However, as we also found semantic similarity to be a modulating factor of whether or not interference can predict facilitation, we will discuss both aspects related to semantic similarity in the General Discussion.

Interestingly, the results also show that natural and man-made items induce different levels of facilitation, with marginally stronger cumulative facilitation for the former than the latter. This is in line with results reported by Belke [30, Exp. 1]. She also found stronger facilitation for natural than for man-made objects but explained that the effect in her study is likely due to a response bias caused by unequal numbers of natural and man-made items (38 and 61%, respectively). In our experiment, the number of man-made and natural objects were perfectly balanced for targets and fillers, thus, there must be another cause underlying the effect. A closer look at the items revealed that natural objects overall had a significantly higher semantic similarity rating within their categories than man-made objects (Mean rating natural: 4.36; Mean rating man-made: 4.55, $t$ = 22.29, $p$ < 0.0001). As this might lead to stronger co-activation between natural objects overall, this is a likely cause for the different levels of facilitation for the two classification types. However, it has also been theorised that the processing of natural and man-made entities substantially differs and that natural entities are somewhat advantaged [71]. Karst and Clapham [72], for example, recently showed that priming is stronger for natural than man-made entities and report that this is partly due to their perceptual properties and familiarity. This might also explain the observed difference in facilitation between the two classification types in our study.

The second analysis of Experiment 2 revealed that cumulative facilitation is independent of the number of intervening items between category members (factor Lag), showing that the longevity attributed to cumulative interference can also be attributed to cumulative facilitation [see also 30]. The fact that we found shorter lags to positively influence overall naming latencies independent of category membership was likely caused by the way the experimental lists were constructed. Short lags mean multiple items of the same classification type (man-made or natural) in close proximity, as all members of a category are of the same classification types. Thus, priming of natural or man-made features is particularly strong, resulting in fast response times of items of the same classification type, independent of their semantic category.

The third analysis of the classification data shows that the cumulative interference effect observed in picture naming can predict the magnitude the cumulative facilitation effect observed in semantic classification of the same targets. This suggests a strong interplay between the two effects and, in turn, supports the claim that both cumulative context effects originate at the conceptual level [9, 30]. As the direction of the effect was rather unexpected and relates to both context effects, we will discuss this aspect further in the General Discussion. However, this leaves open the question why the interference could only predict the facilitation of targets named earlier with simple nouns but not those named with compound words. Please

note: As we initially found this result rather surprising, we ran post-hoc correlation analyses of the data (Spearman's rank correlation as our data was not normally distributed) just to confirm the results of the linear-mixed model. And indeed, we found a significant positive correlation of cumulative interference and cumulative facilitation per subject for simple nouns ($r = 0.50$, $p = 0.002$) but not for compounds ($r = -0.05$, $p = 0.78$). It has been reported that participants prefer categorising and naming objects at the basic level (corresponding to our simple noun targets), and that object classification is faster at the basic level than at subordinate level (corresponding to our compound targets) or superordinate levels [e.g., 65, 66, 73]. Furthermore, it has been argued that the semantic classification task does not necessitate deep, fine-grained semantic processing of information that helps distinguish one category member from the other, or a basic level concept (table) from a subordinate level one (kitchen table), as all category members are either natural or man-made [e.g., 74]. Thus, one possible explanation for our result pattern is that participants processed the depicted objects at the basic level (simple noun concept, e.g., *table* instead of *kitchen table*) in the classification task, independent of how targets had been previously named. As the facilitation effect within categories would then be the result of co-activated basic-level concepts only (*table*, *shelf*, *bed* . . .), its effect size might only be predicted by the activation pattern of the exact same concepts in the picture naming task, namely by basic-level, simple noun names (*table*, *shelf*, *bed*) but not by compound names (*kitchen table*, *bookshelf*, *canopy bed*). However, further research is necessary to confirm this hypothesis.

## General discussion

The aim of the current study was to gain a more comprehensive understanding of cumulative semantic interference. Experiment 1 was a continuous picture naming task designed to investigate whether the magnitude of the effect differs for morphologically complex noun-noun compounds (*kitchen table*, *bookshelf*) and morphologically simple nouns (*table*, *shelf*). The results are clear-cut: While compounds have overall longer naming latencies than simple nouns, both word types induce identical levels of cumulative interference. This study thus provides first evidence that cumulative interference is not affected by morphological complexity. While we predicted that the interference might be weaker for compounds than for simple nouns due to their potentially more complex lemma structure and the fact that the first constituent is not semantically related to the compound's semantic category, the results suggest that the co-activation of semantically related concepts, and thus the interference effect, was mainly driven by the compounds' semantically related second constituents. Any activation that might have dissipated via the unrelated first constituent did not significantly weaken the co-activation of related concepts. As our data thus show that cumulative interference is not affected by morphological complexity.

Please note that we used identical picture stimuli in both word type conditions to minimise potential influences of visual effects on naming times [75]. Thus, one could assume that the identical results in both word type conditions could also be the result of identical visual input leading to the activation of the same conceptual information in both conditions, and thus identical interference. This, however, could not have been the case, as participants would have otherwise produced the same output. After all, to either produce a noun-noun compound or a simple noun, the corresponding conceptual information and lemma(s) for each word type needed to be activated [6, 8, 9 and many others].

While we initially conducted this experiment to investigate the influence of morphological complexity on cumulative interference, the results might also be indicative of the location of the learning mechanism responsible for the effect (i.e., its origin). If it was located at the

interface of the conceptual and the lexical (i.e., lemma) level, one might expect that the lemma representation of the targets influences cumulative interference. More specifically, as we assume different representational formats for compounds and simple nouns [27, 46, 47 but see 43], different magnitudes of cumulative interference should have been observed. This is because in the simple noun condition, the learning mechanism responsible for cumulative interference would only strengthen the connection between the conceptual representation (SHELF) and the one corresponding lemma representation [*shelf*; 22] and potentially weaken the connections to other related targets [29]. In the compound condition, however, multiple lemmas are involved in the production process, the holistic compound lemma (*bookshelf*) as well as the constituent lemmas [*book* and *shelf*; 27, 46]. Thus, the learning mechanism might not only affect the links between the conceptual representation (BOOKSHELF) and the corresponding holistic lemma (*bookshelf*) but also the direct links between the holistic lemma (*bookshelf*) and the constituent lemmas (*book* and *shelf*), and possibly even the links between the constituent lemmas *(book* and *shelf)* and their conceptual representation (BOOK and SHELF). This might affect the activation pattern during compound naming and thus impact on the magnitude of interference that accumulates within categories. If, however, the learning mechanism responsible for cumulative interference was located at the conceptual level itself, implemented as strengthened links between the lexical concept and its features [9, 30], we would expect similar patterns of cumulative interference in both word type conditions. Although the compounds represent more specific concepts compared to the simple nouns, the main features corresponding to a certain semantic category (e.g., furniture: non-living, is wooden, part of a house . . .) are identical for both (SHELF and BOOKSHELF). Thus, a learning mechanism at this level is likely to induce similar cumulative interference for both word types, which is what we observed. This, however, is a post-hoc interpretation of the results and none of the above-mentioned models actually addresses this issue. Nonetheless, this discussion might provide inspiration to consider lexical information of the targets to further investigate the origin of cumulative interference.

Experiment 2 was designed to gain a more comprehensive understanding of the cumulative facilitation found in semantic classification tasks and thus to contribute further to the discussion about the functional origin of cumulative context effects. In this study, we observed the expected cumulative facilitation reflected by a systematic decrease in response latencies within semantic categories, replicating the effect first reported by Belke [30]. We thus provided additional evidence that facilitatory cumulative effects can arise in purely semantic tasks that do not necessarily involve the lexical system. Furthermore, our additional analysis showed that the size of the interference effect found in picture naming can be used as predictor for the facilitation effect of the (simple noun) targets in the classification task, which suggests that both context effects are functionally linked. This replicates previous findings which showed that cumulative interference and cumulative facilitation can influence one another when picture naming and picture classification alternate within an experiment [30, Exp. 5]. Our results thus seem to support accounts that localise the functional origin of cumulative interference at the conceptual level [9, 30]. Models locating the origin at the interface between the conceptual and lexical level [22, 29] can, in their present state, not easily account for cumulative facilitation, as their incremental learning mechanisms responsible for cumulative effects necessarily involves the activation of lexical information. In a comment, however, Oppenheim argued that the model presented in [29] could be altered for the classification task, namely that the learning mechanism affects links between man-made/natural features and man-made/natural response nodes instead of the links between the concepts and the lexical nodes [see 30, p.253]. However, only a computational implementation of the proposed changes can show whether this would indeed result in the observed result pattern found in the classification task as well as the

interplay between interference and facilitation observed in this study. Howard et al. [22] would have to adjust their model even further. In its current form, the proposed learning mechanism only comes into effect when a lexical representation has been selected, which is very unlikely the case in a purely semantic classification task. Thus, they would have to adjust their learning mechanisms in such a way that it does not require actual retrieval but only the activation of lexical information [30; for experimental evidence along those lines, see 76]. This, of course, would only explain cumulative facilitation if the lexical level was indeed activated during the classification task.

While the discussion thus far suggests straightforward results that are more easily explained by models assuming a purely conceptual origin of cumulative interference than those assuming an origin at the interface between the conceptual and lexical level, two findings complicate matters. First, our results show that cumulative facilitation is more strongly affected by repetitions than cumulative interference. While one could argue that this suggests different underlying mechanisms of the two effects which might advocate against a common origin of both effects, we believe that the differences are task specific. It is likely that weak cumulative facilitation after the first classification cycle is due to a ceiling effect in the activation of category-related nodes induced by the persistent activation of man-made/natural features, meaning it is related to the task at hand. After the first classification cycle, the connections between man-made/natural features and all items should have been strengthened [30]. In the subsequent classification cycles, each man-made/natural entity would thus receive some activation when an item of the same type (either man-made or natural) is being classified, even when they do not belong to the same semantic category. This constant activation might exceed any additional activation that the items receive from categorically related items. This would result in overall faster classification times but only a weak accumulation of facilitation within categories, which is what we observed in Presentations 2 to 5. In addition, the general facilitation induced by repetition priming we found in our analysis (main effect for Presentation) adds to the overall enhanced activation levels in subsequent classification cycles, further adding to the ceiling effect. To avoid the constant activation of the same features and to make the classification task and the naming task more similar, one would have to alter the classification task in such a way that participants are required to classify the items into a larger number of classification categories. And even then, general repetition priming might dampen the facilitation effect within categories still more than the interference effect in the naming task, as the effect size of the former is much smaller than that of the latter. However, there might be another task and order-related reason for the strong influence of repetition on cumulative facilitation, as pointed out by Eva Belke during the review process. We initially thought that the decrease of facilitation after the first classification cycle cannot simply be attributed to the fact that activation levels at the conceptual level were exhausted because participants were exposed to the visual stimuli too many times. This was because participants completed an additional round of continuous naming (one naming cycle) after the classification task, and the results mirrored those of the main naming task: robust cumulative interference that did not differ between the two word types (for more details, see S2 Table and S3 Fig in S2 Appendix). While this additional naming cycle at the end of the experimental session is not of central importance for this study, we initially interpreted it as evidence that the stimuli were still sensitive to semantic context effects, even after multiple repetitions. However, it is possible that participants became increasingly aware of the semantic categories due to repeatedly seeing the same items. This is unlikely to have a significant impact on picture naming, as each target still needs fine-grained conceptual processing to distinguish it from other category members, resulting in robust cumulative interference even after being exposed to the targets several times. However, it might have impacted the classification of targets by creating expectations. This might have let to even more superficial processing of the targets and

thus to a decreasing co-activation of semantically related concepts. So, while the influence of repetition clearly differs for cumulative interference and cumulative facilitation, it is likely due to be task-inherent difference rather than inherently different characteristics of the two effects. This, of course, needs to be confirmed by future research.

A second finding that is, at first glance, not easily embedded in the rationale of existing models is that strong cumulative interference in picture naming predicts weak facilitation of simple nouns in the classification task. While this suggests a functional link between the two, the direction of the effect is rather surprising. Please note that we were able to replicate this pattern in a follow-up study. Based on the working model proposed by Belke [30], we initially would have predicted mirroring effects, namely stronger interference predicting stronger facilitation. However, these expectations were built on the rationale that the effect in the naming task was based on conceptual facilitation that turned into interference at the lexical level, and that this facilitation would be identical to the facilitation found in the classification task. In hindsight this may be an oversimplification of the actual processes and there might be multiple explanations for the observed pattern.

First, our initial prediction did not take into account the different tasks that bring to bear the two effects. Assuming a conceptual origin of the effects [30], the accumulating interference in picture naming results from the co-activation of category members and the strengthening of the connections between those members and their semantic features after successful naming. Here, deep processing of all semantic features is key to perform the task. Only if all features are activated, in particular those that are unique to the target and distinguish it from their category members, can one produce the intended category member (see also 30]. In the classification task, on the other hand, only those features are relevant that identify the target as either man-made or natural, which are shared by all members of a category. As argued in the discussion of Experiment 2, fine-grained semantic processing is not essential to perform the classification task [e.g., 30, 70], and quite possibly even disadvantageous as it may delay the response. Thus, the facilitation observed in the classification task is not identical to the facilitation responsible for the interference in the naming task, which might explain why our initial predictions did not pan out.

Secondly, it is possible that the order in which participants completed the two tasks directly influenced the results of the classification task, both with respect to how cumulative facilitation is predicted by the interference effect as well as how it is influenced by semantic similarity. In the naming task, focusing on the unique features of a target (i.e., those it does not share with other category members) is key to complete the task, namely to select one specific target from among a group of (co)activated items. After successfully naming the target, the links between its features and its lexical representation would be strengthened (i.e., learning mechanism). In picture naming, this mechanism leads to the observed interference, as the strengthened links render the target a strong competitor in the naming process of a to-be-named category member. In the following classification task, however, these strengthened unique links will make it increasingly difficult to focus on the shared features of the target, which are key to complete the classification task. Thus, categories that induced strong interference would then induce weaker facilitation than if the target had not previously been named (and no unique features had previously been strengthened). This would also explain why we observed weaker cumulative facilitation for more strongly related category members (i.e., high semantic similarity) in Experiment 2. This rationale is based on results reported by Belke [30]: In Experiment 5, semantic classification and picture naming were mixed, and in the classification task attenuated facilitation was observed after an item was named. Belke argues that this was due strengthening of the unique features after naming, which attenuated the facilitation effect in the classification task, as this is based on the shared features of the category members. However,

until there is a computational implementation of the working model proposed by Belke [30], the above explanation of the results remains speculation.

## Conclusion

The aim of the current study was to contribute to a more comprehensive understanding of cumulative context effects found in picture naming and picture classification. From two experiments, we reported three main findings: 1. Cumulative interference does not significantly differ for morphologically complex noun-noun compounds and morphological simple nouns, suggesting that the effect is not influenced by the morphological complexity. 2. We replicated previous findings that cumulative effects can also be found in purely conceptual tasks, expressed as cumulative facilitation, and 3. we showed that cumulative interference can be used to predict cumulative facilitation. Our results thus indicate a purely conceptual-semantic origin of cumulative context effects, including the much-debated cumulative interference.

## Supporting information

**S1 Appendix.**
(DOCX)

**S2 Appendix.**
(DOCX)

## Acknowledgments

We are grateful to Nura Völk and Mikhail Lomakin for their assistance in data collection and to Guido Kiecker for technical support. We would also like to thank Eva Belke for taking the time to discuss our data prior to submission, and thank her as well as a second anonymous reviewer for their valuable feedback on a previous version of the manuscript.

## Author Contributions

**Conceptualization:** Anna-Lisa Döring, Rasha Abdel Rahman, Pienie Zwitserlood, Antje Lorenz.

**Data curation:** Anna-Lisa Döring.

**Formal analysis:** Anna-Lisa Döring.

**Funding acquisition:** Antje Lorenz.

**Investigation:** Anna-Lisa Döring.

**Methodology:** Anna-Lisa Döring, Rasha Abdel Rahman, Pienie Zwitserlood, Antje Lorenz.

**Project administration:** Anna-Lisa Döring, Antje Lorenz.

**Resources:** Rasha Abdel Rahman, Antje Lorenz.

**Supervision:** Rasha Abdel Rahman, Antje Lorenz.

**Validation:** Anna-Lisa Döring.

**Visualization:** Anna-Lisa Döring.

**Writing – original draft:** Anna-Lisa Döring.

**Writing – review & editing:** Rasha Abdel Rahman, Pienie Zwitserlood, Antje Lorenz.

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
