## [Decision Letter · Decision Letter 0]

30 Dec 2021

PONE-D-21-37276Cumulative semantic interference is blind to morphological complexity and originates at the conceptual levelPLOS ONE

Dear Dr. Döring,

Thank you for submitting your manuscript to PLOS ONE. After careful consideration, we feel that it has merit but does not fully meet PLOS ONE’s publication criteria as it currently stands. Therefore, we invite you to submit a revised version of the manuscript that addresses the points raised during the review process.

 Please submit your revised manuscript by Feb 13 2022 11:59PM. If you will need more time than this to complete your revisions, please reply to this message or contact the journal office at plosone@plos.org. Please include the following items when submitting your revised manuscript:A rebuttal letter that responds to each point raised by the academic editor and reviewer(s). You should upload this letter as a separate file labeled 'Response to Reviewers'.A marked-up copy of your manuscript that highlights changes made to the original version. You should upload this as a separate file labeled 'Revised Manuscript with Track Changes'.An unmarked version of your revised paper without tracked changes. You should upload this as a separate file labeled 'Manuscript'.

We look forward to receiving your revised manuscript.

Kind regards,

Stephanie Ries-Cornou, Ph.D

Academic Editor

PLOS ONE

Journal Requirements:

2. In the ethics statement in the Methods and online submission information, please ensure that you have specified what type you obtained (for instance, written or verbal, and if verbal, how it was documented and witnessed). If your study included minors, state whether you obtained consent from parents or guardians. If the need for consent was waived by the ethics committee, please include this information.

(This research was supported by the German Research Council (LO 2182/1-2 – granted to A.L.). 

The funders had no role in study design, data collection and analysis, decision to publish, or preparation of the manuscript.)

Additional Editor Comments :

Dear authors,

Thank you for submitting your work to Plos One. Please find below detailed evaluations from 2 reviewers. I invite you to respond to each of these comments in detail and to submit a revised manuscript. In particular, I agree with both reviewers that the links with prior work from this team and others (see in particular the comments pertaining to Belke, 2013) should be more clearly detailed and any discrepancy in findings should be interpreted in more details. Please take into account the other comments of course.

Reviewers' comments:

Reviewer's Responses to Questions

**Comments to the Author**

1. Is the manuscript technically sound, and do the data support the conclusions?

Reviewer #1: Partly

Reviewer #2: Yes

2. Has the statistical analysis been performed appropriately and rigorously? 

Reviewer #1: Yes

Reviewer #2: Yes

3. Have the authors made all data underlying the findings in their manuscript fully available?

Reviewer #1: No

Reviewer #2: No

4. Is the manuscript presented in an intelligible fashion and written in standard English?

Reviewer #1: Yes

Reviewer #2: Yes

5. Review Comments to the Author

Reviewer #1: In this paper, Döring and colleagues report a study investigating whether morphologically complex noun-noun compounds elicit cumulative semantic interference (CSI) in continuous naming and cumulative semantic facilitation/priming in man-made/natural decisions similarly to simple nouns. In both experiments, results were similar to those observed for simple nouns.

As the authors themselves note, given that previous studies with simple nouns (e.g., Belke, 2013; Riley et al., 2014) have shown the origin of the CSI effect is at the conceptual level and its locus at the lexical level, one could argue that morphology is largely irrelevant to the argument concerning its origin but potentially relevant to its lexical locus. However, I’m not convinced the experiments do provide a straightforward test of whether activation patterns of compounds differ from simple nouns, particularly as some discrepant results need further explanation.

Major concerns

The motivation for conducting the study emphasises different patterns of CSI might arise due to the single vs. multiple lemma account of compounds. The authors’ preferred hypothesis is that compounds will elicit weaker CSI as constituent lemmas become activated, including those of the unrelated modifiers that will essentially add noise to the system. The authors seem to have previously addressed the multiple lemma issue in their recent publication that they cite but don’t discuss in detail when motivating the present study (see doi link below). That study showed the constituent lemmas of compounds are activated and that significantly weaker CSI occurs for compounds than simple nouns – as predicted in the present study. So, is the finding of no difference in magnitude of CSI between compounds and simple nouns in Experiment 1 then a failed replication of this previous result? If so, this should be addressed squarely in both the Introduction and Discussion.

Döring, A.-L., Abdel Rahman, R., Zwitserlood, P., & Lorenz, A. (2021). On the lexical representation(s) of compounds: A continuous picture naming study. Journal of Experimental Psychology: Learning, Memory, and Cognition. Advance online publication. https://doi.org/10.1037/xlm0001049

Aren’t compound words much lower in lexical frequency than simple nouns? For example, according to the SUBTLEXus norms, the frequencies for ‘shelf’ and ‘bookshelf’ are 6.96 vs. 0.33, respectively. Starreveld et al. (2013) reported an interaction between lexical frequency and semantic relatedness in the picture-word interference task, with a smaller interference effect for high frequency distractors. Might not a similar interaction occur in continuous naming, so that CSI for simple nouns would be expected to be smaller? One way of addressing this would be to include lexical frequencies for both word types in the analyses.

Starreveld, P. A., La Heij, W., & Verdonschot, R. (2013). Time course analysis of the effects of distractor frequency and categorical relatedness in picture naming: An evaluation of the response exclusion account. Language and Cognitive Processes, 28(5), 633–654.

On p. 24, the authors state that CSI differed for natural and man-made objects, but this analysis is not reported in the Results section for Experiment 1. More importantly, Belke (2013) did not report stronger facilitation for natural than man-made objects during superordinate classification. Rather she reported the opposite: participants were faster to classify man-made objects. Riley et al. (2014) also reported that man-made objects were classified more quickly. Hence, the results from Experiment 2 are also not consistent with prior work. In the object recognition literature, faster superordinate classifications observed for man-made objects are typically attributed to them having fewer features than living/natural objects (see Clark & Tyler, 2015). Is it possible that the manmade stimuli employed here actually contain more features and so took longer to process (e.g., does the picture of a bookshelf also show books, making it more complex than just a picture of an empty shelf)?

Clarke, A., & Tyler, L. K. (2015). Understanding what we see: How we derive meaning from vision. Trends in Cognitive Science, 19, 677-687.

The unexpected finding that stronger cumulative facilitation was associated with weaker interference, but only for simple nouns, also requires some more explanation. As the classification task was always run after the naming task, weaker interference predicts stronger facilitation, not the other way round. Participants may simply have benefited from the repeated exposure to the target concepts. Note that Belke (2013) counterbalanced her order of naming and classification tasks.

Minor concerns

One methodological difference also precludes a clearcut interpretation. In the typical continuous naming paradigm, there is no familiarisation phase. Here, the authors employ a familiarisation phase to ensure the correct compound or simple noun was produced. In the picture-word interference paradigm, inclusion of a familiarisation phase has been reported to influence the direction and magnitude of semantic effects (e.g., Collina et al., 2013; Gauvin et al., 2018). What effects on conceptual and/or lemma activation levels might inclusion of this familiarisation phase have produced in the continuous paradigm? Belke (2013) introduced a familiarisation phase but Riley et al. (2014) did not, and both reported identical results for semantic classification, suggesting this factor might not be important for CSI with frequently used, simple nouns.

However, familiarisation might have affected the activation levels of the compounds, if they are less frequently used (see comments above, and Miozzo & Caramazza, 2003 for evidence that familiarising participants with low frequency words reduces the magnitude of the frequency effect).

Collina, S., Tabossi, P., & De Simone, F. (2013). Word production and the picture-word interference paradigm: The role of learning. Journal of Psycholinguistic Research, 42, 461-473.

Gauvin, H. S., Jonen, M. K., Choi, J., McMahon, K., & de Zubicaray, G. I. (2018). No lexical competition without priming: Evidence from the picture-word interference paradigm. Quarterly Journal of Experimental Psychology, 7, 2562-2570.

Miozzo, M., & Caramazza, A. (2003). When more is less: A counterintuitive effect of distractor frequency in the picture-word interference paradigm. Journal of Experimental Psychology: General, 132(2), 228–252.

Did lag or serial position influence any of the results? On p. 12 lag is manipulated by 2, 4, 6, or 8 items but the results section doesn’t mention if lag was included as a covariate in any of the models. Although lags < 8 shouldn’t result in different CSI (see Schnur, 2012), it is still important to check. Similarly, linear effects due to serial order/time on task should be checked per Howard et al. (2006).

Schnur, T. T. (2014). The persistence of cumulative semantic interference during naming. Journal of Memory and Language, 75, 27–44.

Reviewer #2: Review of "Cumulative semantic interference is blind to morphological complexity and originates at the conceptual level"

Eva Belke

The authors report the results from a three-part continuous naming and classification experiment requiring participants to first name a continuous item list four times, then classify it four times and finally name it once again. Participants named half of the items with more specific compound names and the other half with simple nouns, which they had been familiarized with prior to the experiment. There were consistent cumulative semantic interference effects in each of the first four naming rounds and the follow-up naming round. They were identical in strength for both simple nouns and compound nouns. The authors take these findings to support the notion that the context effect originates at the conceptual level and comes to effect at the lemma level. An increase in the semantic similarity of the category exemplars led to an increase the semantic interference effect in naming in all four rounds. In semantic categorization, it led to an increase in classification in the first presentation but none of the repetitions, which did not yield any context effect at all.

In a combined analysis of the first two sections of the experiment, the authors used the magnitude of the facilitation effect per category and repetition as a predictor in modelling the reaction times in the naming task. They found that stronger facilitation predicted weaker interference, which seems to run counter to the predictions I have made in Belke (2013). The authors discuss this finding in light of differences between the two tasks and argue that it might be compatible with the notion that facilitation and interference cancel each other out, as predicted by the swinging lexical network account.

As the authors note, I have discussed their data with them before, and reading the paper has been a good opportunity to think about the data again based on all the details of the experimental design. The paper is written well and the data should be published in my view. However, I have a few issues with the data analyses and the discussion of the data in the paper that need to be addressed in my view. I would think that all of them can be addressed in a minor revision.

1. The correlation of facilitation and interference.

In the supplementary analyses, the authors aim at predicting the context effect in naming from the context effect in classification and find that weaker classification effects yield stronger interference effects. At first sight, this seems to be incompatible with the account put forward in Belke (2013) and, as I will argue shortly, with the Swinging Lexical Network (SLN) account, too. However, there are two points to keep in mind when reasoning about the data: First, the classification task has different task requirements to those of the naming task, as the authors note in the General Discussion. Accordingly, the kind of facilitation arising in semantic classification is different from that arising in naming, which we cannot access directly. Second, reading up on the details of the design I realized that all participants first named the items and then classified them. If we assume that the context effect in naming and classification is mediated by some form of conceptual learning at the interface of lexical concepts and conceptual features, the information that is learnt is bound to differ between naming and classification due to the different task affordances. While in naming, unique features of the category exemplars are very relevant to performing the task, the classification task is more reliant on features shared across items of a set. Accordingly, the conceptual facilitation seen in semantic classification is not identical to the conceptual facilitation that arises as part of the naming process, and the order in which the tasks are administered matters (see Experiment 5 in Belke,2013) and is a relevant piece of information in accounting for the present data. Indeed, when comparing the non-deviant trials shared between Experiments 1 and 5 in Belke (2013), I found that the semantic facilitation effect was attenuated when the third or fourth item in a set was named (Experiment 5) rather than classified (Experiment 1). Arguably, this was due to the fact that naming required a focus on the non-shared information, which attenuated the facilitation effect on classification.

Applying this reasoning to the present data, it is quite plausible that strong interference effects in naming yield weak facilitation effects in classification and vice versa. Naming items from closely related sets requires speakers to focus on the unique features of a concept, and the retuning of the relevant links arising from this is likely to render the representations of same-category members functionally less similar to each other. Accordingly, the semantic facilitation effect will be less strong than for items that have not been rendered more dissimilar before (by virtue of sharing fewer common features in the first place).

One implication of this is that in the present design, the causal relation between the effects seen in naming and classification actually works in the reverse direction than that implemented in the analyses, that is, if anything, the naming data predict the classification data rather than vice versa. Using the classification times as predictor of the naming times might therefore not be appropriate. I was wondering whether a plain correlation between the predicted slopes of the effect in naming and categorization would be a more appropriate way of analyzing the data. If the authors used slopes generated from the raw data, they may want to run a partial correlation that takes into account effects of semantic similarity.

Either way, I would strongly suggest that the authors work with the context effect/the slope of the first presentation in the classification task only. It is clear that after that, all that needs to be learnt to perform that task has been learnt and RTs reach floor, so the context effects/the slopes for the repetitions are even less valid predictors of the conceptual facilitation seen in naming.

Let me conclude this point by pointing out that, despite the evidence in favor of the reasoning outlined above, it remains somewhat speculative until we have an implementation of a working model at our disposal that can simulate conceptual learning in naming or classification tasks.

2. Conceptual accumulation account vs. (?) SLN?

Related to the previous point, I wonder whether the predictions from the account I have put forward in Belke (2013) and those from the SLN, augmented by a learning mechanism, would be different in any way. After all, the accounts are both based on WEAVER++ as a reference model and both accounts predict that there is conceptual facilitation and lexical interference in naming, with lexical interference overriding conceptual facilitation in the continuous naming paradigm. According to the conceptual accumulation account, the an increase in conceptual facilitation causes a corresponding increase in lexical interference, accounting for why, in net reaction times, there is a continuous linear increase with each item that has been named. The SLN, once augmented by a learning mechanism, would make the same predictions, in my reading.

3. Semantic similarity effects in classification.

In the analysis of the data from Experiment 2, semantic similarity is included in a two-way and a three-way interaction but not as a main effect. From all I know about modelling interaction effects, the corresponding main effects should always be included as well, unless there is very good reason to exclude them. I cannot see such reasons here, especially since the main effect was included in the model of the data of Experiment 1.

Once the main effect is included, it would be interesting to see how this impacts on the interaction of ordinal position and semantic similarity. In the model reported in the paper, this interaction does not reach significance (t < 2 and p = .089) but there is a trend that category exemplars that are more similar to their category neighbors yield LESS facilitation rather than more, which would be compatible with the rationale I have laid out in 1.

The semantic similarity : ordinal position interaction further interacts with repetition, and the authors argue that the ordinal position effect was stronger initially for categories with very similar exemplars as compared to categories with less similar exemplars. This does not seem compatible with the trend seen for the two-way interaction; if the authors were right, we should see a *negative* trend in the two-way interaction (ordinal position : semantic similarity) rather than a positive one, if I am not mistaken. Could the authors maybe plot the three-way-interaction? My impression from reading the model was that the two-way interaction (less facilitation for more similar category exemplars) vanishes after the first repetition and that this led to the three-way-interaction.

4. Dissipation of the semantic facilitation effect in Experiment 2.

I have argued above that the facilitation effect in classification dissipates because the information that is learnt in the first round of classification renders the task so easy that the facilitation effect is not discernable any more, or non-existent. I was wondering whether in addition to that, participants may have become aware of the shared category membership of the items in the lists over time. It did not become entirely clear to me what the order of the familiarization and naming tasks was like, given that the lists were tested blocked by word type. Is it correct that in Experiment 1, participants first received the familiarization for the first set of 60 items, then named it four times, then received the second familiarization and subsequently named the second set of items four times? In that case, it would probably be quite likely that participants became aware of the subsets included in the lists. Schnur, Roelofs and others have discussed that the design as used by the authors (including a lag-2 condition) may raise participants' awareness of the item sets, and this is even more likely to be the case when the list included only relatively few items. Awareness of the sets is probably of little use for the naming task (as participants still have to individuate conceptually each concept from its category coordinates, see Roelofs, 2018), but it is likely that it impacted on the categorization task. Hence, the fact that the context effect re-appeared in original strength in the naming phase after the categorization task (see p. 24) does not refute this reading of the data, as this phase employed a fundamentally different task for which the superordinate category information alone was not sufficient (see also point 1.).

5. Non-interaction of word type and ordinal position.

The authors find no interaction of word type and ordinal position once the effect of semantic similarity is accounted for. I share the authors' view that this provides evidence for a conceptual origin of the context effect. However, one might argue that the intensive familiarization may have turned a genuine naming task into some form of cued recall task that might dampen constituent effects on the selection of the compound name at the lemma level. Could that be the case, or do compound production experiments always include a familiarization comparable to that used in the present experiment? If so, does this also hold for experiments that provide evidence for lexical competition between the compound and its constituents?

6. In all models reported in the paper, categories and items are fitted as separate random effects even though they are probably highly correlated. Why did the authors not code the random effects as a nested structure (i. e. 1 + word type | category/item)? This seems more true to the design to me.

7. Further to this point, my impression is that some of the predictors are correlated to some extent. Most notably, in the models reported in Tables S2 and S3, the slope of the facilitation effect (i. e., the difference between the RTs at ordinal position 5 and 1) and semantic similarity are included as separate factors, but the facilitation slope is most likely influenced by semantic similarity, as is the interaction between the two factors. It may be useful to assess to what extent such collinearity impacts on the models, not only for models S2 and S3 but all models.

Minor points.

The data have not yet been made available but the authors have announced that they will be made available in an OSF repository.

On p. 4, the authors may want to make the distinction between the terms origin and locus more explicit. It is there but may be easily missed by those who have not read about it before.

On p. 9, I am not sure I can follow the argumentation about the effect co-activated first constituents might have. Does the conceptual co-activation of the unrelated first constituent really reduce the activation in the network pertaining to the concept of the target word? Is there evidence that suggests this might be the case? Would the activation in the cohort not be primary, i. e. direct, and the activation in the first constituents indirect, mediated via the lemma level?

p. 30, hindside -> hindsight

6. PLOS authors have the option to publish the peer review history of their article (what does this mean?). If published, this will include your full peer review and any attached files.

Reviewer #1: No

Reviewer #2: **Yes: **Eva Belke

---

## [Author Response · Author response to Decision Letter 0]

18 Feb 2022

Please find a detailed response to all reviewer comments in the attached document "Responses to Reviewers"

---

## [Decision Letter · Decision Letter 1]

20 Apr 2022

PONE-D-21-37276R1Cumulative semantic interference is blind to morphological complexity and originates at the conceptual levelPLOS ONE

Dear Dr. Döring,

Thank you for submitting your manuscript to PLOS ONE. After careful consideration, we feel that it has merit but does not fully meet PLOS ONE’s publication criteria as it currently stands. Therefore, we invite you to submit a revised version of the manuscript that addresses the points raised during the review process.

Dear authors,

Congratulations on your excellent work on this revision! Both reviewers were highly satisfied. The only thing remaining is to correct the small misunderstanding pointed out by Reviewer 2 (Eva Belke) in the final version of your manuscript. I will not send it out again for review though and I anticipate being able to accept your manuscript for publication upon receipt.

Congratulations again!

We look forward to receiving your revised manuscript.

Kind regards,

Stephanie Ries-Cornou, Ph.D

Academic Editor

PLOS ONE

Journal Requirements:

Reviewers' comments:

Reviewer's Responses to Questions

**Comments to the Author**

1. If the authors have adequately addressed your comments raised in a previous round of review and you feel that this manuscript is now acceptable for publication, you may indicate that here to bypass the “Comments to the Author” section, enter your conflict of interest statement in the “Confidential to Editor” section, and submit your "Accept" recommendation.

Reviewer #1: All comments have been addressed

Reviewer #2: All comments have been addressed

2. Is the manuscript technically sound, and do the data support the conclusions?

Reviewer #1: Yes

Reviewer #2: Yes

3. Has the statistical analysis been performed appropriately and rigorously? 

Reviewer #1: Yes

Reviewer #2: Yes

4. Have the authors made all data underlying the findings in their manuscript fully available?

Reviewer #1: Yes

Reviewer #2: Yes

5. Is the manuscript presented in an intelligible fashion and written in standard English?

Reviewer #1: Yes

Reviewer #2: Yes

6. Review Comments to the Author

Reviewer #1: The authors have address all of my concerns. The novel findings from the additional analyses will make an interesting contribution to the literature.

Reviewer #2: I am happy with the revisions, except for one point that I would ask the authors to revise. The authors have picked up and addressed my ideas about how the order of tasks may have affected the results, which I appreciate. On p. 34, they note that:

"While this outlined mechanism accounts for the bulk of our data, it might not be entirely compatible with the finding that cumulative interference robustly survives multiple repetitions. If category members are rendered less similar after naming, this should result in weaker co-activation between category members after the first naming cycle. Repeating the items should, in turn, be associated with less cumulative interference. This is, however, not what we observed [see also 27,28,34]."

This is incorrect, and I can see in hindsight that the way I phrased my ideas in my original review may have been too sloppy and, as a result, misleading:

I have not argued that links to shared features are being weakened after naming; after all, they are typically central to the concept and hence needed for it to be activated and named, even if they are not distinctive. Instead, I have argued that less central, individualizing features are being strengthened after naming, as speakers have to focus on these features when trying to resolve the competition between the target and its competitors.

A target concept that has been named before will be co-activated on the next trial, as will be its individualizing features, rendering it harder for the speaker to focus on the individualizing features of the current target. The more targets of a category have been named, the more such individualizing features will be co-activated, causing the previous targets to be strong competitors and rendering target naming increasingly hard. If this was not predicted by the account, it would not be able to account for the cumulative context effect in the first place.

Strengthening individualizing features of a target also causes repetition priming on the next run through the experiment, so the account can explain both effects. I have not found any interaction of the context effect with repetition/load either in Belke (2013, Experiment 1; see FN 3) and I do not think that this finding is incompatible with the account I put forward in the review.

Unlike naming, classification requires that speakers focus on the shared features, which causes the concept-feature links to these features to be strengthened and leads to cumulative facilitation as well as repetition priming.

I would be grateful if the authors could clarify this point in their final version of the paper. Labeling the strengthening of individualizing features as targets becoming less similar is misleading, and it would be good if the authors revised it accordingly.

7. PLOS authors have the option to publish the peer review history of their article (what does this mean?). If published, this will include your full peer review and any attached files.

Reviewer #1: No

Reviewer #2: **Yes: **Eva Belke

---

## [Author Response · Author response to Decision Letter 1]

27 Apr 2022

Our response to the reviewers can be found in the separate file called "Response to reviewers".

---

## [Editor Report · Decision Letter 2]

11 May 2022

Cumulative semantic interference is blind to morphological complexity and originates at the conceptual level

PONE-D-21-37276R2

Dear Dr. Döring,

Congratulations!! We’re pleased to inform you that your manuscript has been judged scientifically suitable for publication and will be formally accepted for publication once it meets all outstanding technical requirements.

Kind regards,

Stephanie Ries-Cornou, Ph.D

Academic Editor

PLOS ONE
---

## [Editor Report · Acceptance letter]

18 May 2022

PONE-D-21-37276R2 

Cumulative semantic interference is blind to morphological complexity and originates at the conceptual level 

Dear Dr. Döring:

I'm pleased to inform you that your manuscript has been deemed suitable for publication in PLOS ONE. Congratulations! Your manuscript is now with our production department. 

Kind regards, 

on behalf of

Dr. Stephanie Ries-Cornou 

Academic Editor

PLOS ONE